# The Impact of Temperature and Ethanol Concentration on the Global Recovery of Specific Polyphenols in an Integrated HPLE/RP Process on Carménère Pomace Extracts

**DOI:** 10.3390/molecules24173145

**Published:** 2019-08-29

**Authors:** Nils Leander Huaman-Castilla, Maximiliano Martínez-Cifuentes, Conrado Camilo, Franco Pedreschi, María Mariotti-Celis, José Ricardo Pérez-Correa

**Affiliations:** 1Chemical and Bioprocess Engineering Department, School of Engineering, Pontificia Universidad Católica de Chile, Vicuña Mackenna 4860, P.O. Box 306, Santiago 7820436, Chile; 2Escuela de Ingeniería Agroindustrial, Universidad Nacional de Moquegua, Prolongación calle Ancash s/n, Moquegua 18001, Peru; 3Centro Integrativo de Biología y Química Aplicada (CIBQA), Escuela de Tecnología Médica, Facultad de Salud, Universidad Bernardo O’Higgins, General Gana 1702, Santiago 8370993, Chile; 4Centro de Aromas y Sabores (DICTUC S.A.), Vicuña Mackenna 4860, P.O. Box 306, Santiago 7820436, Chile; 5Programa Institucional de Fomento a la Investigación, Desarrollo e Innovación, Universidad Tecnológica Metropolitana, Ignacio Valdivieso 2409, P.O. Box 9845, Santiago 8940577, Chile

**Keywords:** polyphenols, ethanol, hot pressurized liquid extraction, resin purification, solvent polarity, selectivity

## Abstract

Sequential extraction and purification stages are required to obtain extracts rich in specific polyphenols. However, both separation processes are often optimized independently and the effect of the integrated process on the global recovery of polyphenols has not been fully elucidated yet. We assessed the impact of hot-pressurized liquid extraction (HPLE) conditions (temperature: 90–150 °C; ethanol concentration: 15%–50%) on the global recovery of specific phenolic acids, flavanols, flavonols and stilbenes from *Carménère* grape pomace in an integrated HPLE/resin purification (RP) process. HPLE of phenolic acids, flavanols and stilbenes were favored when temperature and ethanol concentration increased, except for chlorogenic acid which showed an increment of its Gibbs free energy of solvation at higher ethanol contents. Ethanol concentration significantly impacted the global yield of the integrated HPLE/RP process. The lower the ethanol content of the HPLE extracts, the higher the recovery of phenolic acids, flavanols and stilbenes after RP, except for flavonols which present more polar functional groups. The best specific recovery conditions were 150 °C and ethanol concentrations of 15%, 32.5% and 50% for phenolic acids, flavanols and stilbenes, and flavonols, respectively. At 150 °C and 32.5% of ethanol, the extracts presented the highest total polyphenol content and antioxidant capacity. The integrated HPLE/RP process allows a selective separation of specific polyphenols and eliminates the interfering compounds, ensuring the safety of the extracts at all evaluated conditions.

## 1. Introduction

Carménère wine production generates ~80.000 t/year of grape pomace (skin and seed residuals) [1]. This agroindustrial waste has low economic value, but is an excellent natural source of polyphenols (18–56 mg gallic acid equivalent (GAE)/g) [2]. Flavanols, flavonols, phenolic acids and stilbenes are the most representative polyphenolic compounds present in Carménère wine pomace [3]. This pomace is rich in flavanols, such as epigallocatechin and catechin, that show a high antioxidant activity and partly explain the distinctive astringency of Carménère wines [3,4]. In addition, Carménère grapes contain large amounts of the flavonol quercetin, which favor the co-pigmentation process in red wines, and have been related to the prevention of obese-related diseases [3,5]. Abundant phenolic acids in these grapes, such as gallic and caffeic acid, have shown bioactivity against skin cancer [6]. Resveratrol is the main stilbene present in the grape’s skins and represent another important group of Carménère polyphenols, which has demonstrated anticancer effects [7,8]. Consequently, efficient and food grade extraction processes to obtain extracts rich in specific polyphenols are desirable to commercially produce functional ingredients for a wide spectrum of food and nutraceutical applications.

Conventional extraction (solid–liquid) uses organic solvents to obtain bioactive extracts from vegetable matrices [9]. Commonly these solvents are chosen according to their dielectric constant (*ε*) that is assumed to be correlated with the solvent polarity [10]. However, the ability of solvents to form hydrogen bonds with the metabolite of interest should also be considered when choosing the right solvent, because both properties significantly impact the solvation capacity of solvents [11,12]. 

Water mixtures with acetone and methanol present intermediate polarity and they are able to form hydrogen bonds. These mixtures have been shown useful to recover a wide range of polyphenols from plant materials [13]. According to the structure of specific polyphenols, some mixtures are more appropriate [14]. For example, a conventional extraction (28 °C) with an acetone solution (65%) is efficient to obtain flavanols; while a methanol solution (85%) under the same operating conditions is more efficient to obtain flavonols [15]. Interestingly, when the temperature of this methanol solution is increased to 50 °C, the extraction is more selective for stilbenes [16]. Consequently, the extractability of polyphenols in conventional extraction depends on many factors, such as the solvation properties of the solvent, the extraction temperature and the polyphenol´s structure [10,12,17].

Nevertheless, conventional extraction processes are slow and consume large volumes of solvents which are usually not food-grade and not friendly to the environment [18]. Hot-pressurized liquid extraction (HPLE) is an alternative food-grade and clean technology that allows reducing the extraction time and solvent volumes as well as improving the extraction yield of polyphenols [17,19]. High pressures keep the solvent in liquid state and favor its diffusion into the plant matrix [19]. Like in conventional extraction, the extractability of polyphenols in HPLE is also influenced by the solvation properties of the solvent, extraction temperature and the polyphenol´s structure [19,20]. As illustrated in the Appendix A, high pressure increases the dielectric constant of pure water, while at elevated temperatures the dielectric constant is reduced [21]. In turn, both pressure and temperature decrease the polarity/polarizability Kamlet–Taft parameter (π*) and the polarity/acidity Reichardt dye (E_T(30)_) parameter for water [22,23]. Since choosing the right solvent to preferentially recover a given polyphenol using HPLE is challenging, experimental measurements can be complemented with computational tools. These have been successfully used for calculating the dielectric constant of solvent mixtures [24,25] and for establishing the molecular interactions between the solvent and the metabolite of interest.

Normally, the content of total polyphenols and antioxidant capacity of the extracts increase with temperature; however, some specific polyphenols could be degraded during HPLE [26]. In grape pomace extraction, pure water at high temperatures (T° ≥ 120 °C) increases the degradation rate of low and high molecular weight polyphenols such as kaempferol, quercetin, catechin, epicatechin and procyanidins [27,28]. In addition, the recovery of reducing sugars and the generation of Maillard undesirable compounds are favored at high temperatures [29], complicating the handling of the extracts in the subsequent operations (e.g., atomization) and the formulation of functional foods and nutraceuticals [30,31].

Previous research studies of HPLE have analyzed the effects of temperature (60–200 °C) and ethanol as co-solvents (0%–100%) in the recovery of total polyphenols, their degradation as well as the generation and recovery of undesirable compounds (hydroxymethyl furfural (HMF) and sugars) [32,33,34]. However, few studies have analyzed the effect of these conditions on the profile of the polyphenols [32,35,36]. These studies show that it is difficult to predict the effects of co-solvent and temperature on the recovery of specific polyphenols; hence, optimal extraction conditions should be obtained by experimental design.

To purify polyphenol extracts, macroporous resin processes are widely used since they are easily scalable and relatively low cost; in addition, resins are reusable and non-toxic [37]. With this technique, unwanted compounds are eliminated and specific polyphenols with a given chemical structure and molecular weight are adsorbed selectively, depending on the resin and the characteristics of the solvent [37,38]. The purification process consists of two stages: adsorption, where some polyphenols are retained in the resin, and desorption where the retained polyphenols are liberated using a water/ethanol solution as eluent [39]. In aqueous extracts of grape pomace, catechin, epicatechin, quercetin and kaempferol were successfully separated with the FPX-66 resin using a 70% ethanol solution as eluent [40]. Sun et al. [41] found that the X5 macroporous resin (70% ethanol solution) allowed the selective separation of chlorogenic acid from aqueous extracts of apple pomace. In previous research, we observed that the use of an ethanol solution (15%) in HPLE reduced the adsorption of total polyphenols in the subsequent purification stage, affecting their global recovery; however, some specific polyphenols like epicatechin (flavanols) and feruloylquinic acid (phenolic acid) increased their global yield [39]. Thus, the ethanol content in HPLE also affects the recovery of total and specific polyphenols during the purification process.

Normally, the extraction stage in HPLE is designed to maximize the recovery of total polyphenols. Then, after applying these optimal extraction conditions, the purification stage is optimized independently to get rid of unwanted compounds [42,43]. Only a few studies have optimized the extraction and purification simultaneously to recover specific compounds [39,41,43]. Previously, we have demonstrated that by using relatively low concentrations of ethanol (15%) at 90 °C in HPLE, the global recovery (after purification) of total polyphenols and of some specific polyphenols of low and high molecular weight is reduced [35,39]. By using quantum chemical calculations, we concluded that the molecular dimensions of some polyphenols did not affect their selectivity in this integrated HPLE/resin purification (RP) process [39]. 

The behavior of specific polyphenols in different solvents should be assessed using both empirical and mathematical methods since they are complementary [44]. In this sense, the implicit solvation models allow us to understand the physicochemical behavior of intermolecular interactions between the metabolite of interest and the solution, where the solvent is represented as a polarizable dielectric continuum [45,46]. These continuum models are frequently used in quantum mechanics calculations [47,48]. Additionally, quantum mechanics, which incorporate the electronic structure, improve the characterization of the polarization of the solute by the solvent, and allow for a more accurate description of the molecular shape and charge distribution [24,25].

In this study, we analyzed the effect of varying HPLE conditions of temperature (90–150 °C) and ethanol concentration (15%–50%) on the global recovery of some specific polyphenols in an integrated HPLE/RP process. Additionally, we applied computational chemical calculations to understand the selective recovery of some polyphenols and the solvent characteristics under HPLE conditions. Hence, we will be able to define optimal extraction conditions depending on the target polyphenol to recover.

## 2. Materials and Methods

Carménère grape pomace was extracted by HPLE with several water-ethanol mixtures and subsequently purified using a water/ethanol solution as desorption eluent. The crude and purified extracts were chemically characterized through (i) specific polyphenols profiles; (ii) fructose and glucose content; and (iii) 5-hydroxymethylfurfural (HMF) concentration.

### 2.1. Chemicals and Analytic Reagents

Ethanol (Sigma Aldrich, St. Louis, USA) was used for both the extraction and purification stages. For the chemical analysis, the following analytical grade reagents were used: Folin–Ciocalteu reagent, Carrez solution I, Carrez solution II, dimethylaminocinnamaldehyde (DMAC; F.W. 175.23), sodium carbonate, sodium chloride, sodium hydroxide, ammonium hydroxide, HMF (Sigma Aldrich, St. Louis, USA). The standards used in the polyphenol profiles, such as gallic acid (≥99%), catechin (≥98%), epigallocatechin (≥98%), epicatechin (≥98%), kaempferol (≥98%), resveratrol (≥98%), quercetin (≥97%), caffeic acid (≥99%), chlorogenic acid (≥98%), vanillic acid (≥99%) and ferulic acid (≥98%) were purchased from Xi’an Haoxuan Bio-Tech Co., Ltd. (Baqiao, China). 

### 2.2. Wine Pomace

Carménère wine pomace was obtained from Concha y Toro Vineyard, Region del Maule, Chile. Wine pomace samples were taken after the Carménère wine process had finished; the samples were then immediately frozen (−20 °C). Subsequently, samples were ground with an Oster blender (Sunbeam Products, Inc., Boca Raton, FL, USA) to a particle size smaller than 1 mm diameter.

### 2.3. Hot-Pressurized Liquid Extraction (HPLE) of Carménère Pomace

This process was developed according to the methodology of Mariotti-Celis et al. [35] with some modifications. Carménère pomace samples of 5 g (dry weight) were extracted at 90, 120 and 150 °C using different amounts of ethanol (15%, 32.5% and 50%) as co-solvents in an accelerated solvent extraction device (ASE 150, Dionex) applying pressurized nitrogen (~10.2 atm). Raw extracts were frozen (−20 °C) in amber vials until the chemical analysis.

### 2.4. Purification Process (RP) of Carménère Pomace Raw Extracts

Carménère pomace raw extracts were purified at 30 °C using a glass column (Ø: 25 mm; h: 100 mm) packed with ~18g of HP-20 resin (Diaion, Tokyo, Japan) according to the methodology of Mariotti-Celis et al. [35] with some modifications. First, 50 mL of raw extract were passed through the resin with a flow rate of 3 mL/min; then, after the column was washed with 100 mL of distilled water, polyphenols were desorbed with a water/ethanol solution (80%) with a flow rate of 3 mL/min. Finally, the column was regenerated using 100 mL of distilled water, 100 mL of NaOH (1N) and 100 mL of HCL (2N).

### 2.5. Total Polyphenols Content (TPC) 

Total polyphenols content (TPC) levels of the Carménère pomace extracts were determined by Folin–Ciocalteu assay [49]. A volume of 3.75 mL of distilled water, 0.5 mL of Carménère pomace extract and 0.25 mL of Folin–Ciocalteu reactive (1N) were mixed with 0.5 mL of a sodium carbonate solution (10% *w*/*v*). Absorbance was measured at 765 nm (Spectrometer UV 1240, Shimadzu, Kioto, Japan) after a reaction time of 1 h at 20 °C. Results were expressed as mg of GAE per gram of dried pomace.

### 2.6. Antioxidant Capacity 

The antioxidant capacity of the extracts was determined using the DPPH (2,2-diphenyl-1-picrylhydrazyl) radical scavenging method [50]. In summary, first 0.1 mL of extract was mixed with 3.9 mL of DPPH solution (0.1 mM), then the solution was mixed by vortex for 10 s and incubated at room temperature in the dark for 30 min. The reduction of DPPH was measured at 517nm (Spectrometer UV 1240, Shimadzu, Kioto, Japan). The IC50 (mg/L), defined as the effective extract concentration needed to inhibit 50% of DPPH radical absorption, was calculated and compared with Trolox, using the Trolox equivalent antioxidant capacity (TEAC) equation (TEAC = IC50 Trolox/IC50 sample) [51]. Antioxidant capacity values were expressed as µM of Trolox equivalent (TE) per gram of dry mass of pomace.

### 2.7. Quantification of 5-Hydroxymethylfurfural (HMF) Concentration

HMF content was determined according to the methodology of Mariotti-Celis et al. [35], using a high-performance-liquid-chromatography with a diode-array detector (HPLC-DAD) (Thermo Scientific Dionex Ultimate 3000, Waltham, MA, USA) equipped with a reverse phase Acclaim TM 120 C18 column. Results were expressed as mg of HMF per g of dry mass of pomace.

### 2.8. Quantification of Fructose and Glucose Concentration

Glucose and fructose concentrations of the extracts were measured according to the methodology of Mariotti-Celis et al. [35]. Analyses were performed by high-performance liquid chromatography (HPLC)–infrared (IR) (Thermo Scientific Dionex Ultimate 3000, Massachusetts, USA) equipped with a normal phase Li ChroCART ^®^ 250-4 Purospher ^®^ STAR (5 µm). Results were expressed as mg of glucose or fructose per g of dry mass of pomace. 

### 2.9. Quantification of Target Polyphenols 

The presence of gallic acid, catechin, epigallocatechin, epicatechin, kaempferol, resveratrol, quercetin, caffeic acid and chlorogenic acid was quantified according to the methodology of Liu et al. [52] with some modifications. Samples of 1 mL were diluted with distillated water (1/10) and filtered through a 0.22 mm membrane. Then, 5 µL of filtered sample was injected into a ultra-performance liquid chromatography–mass spectrometry (UPLC–MS, Dionex Ultimate 3000 with Detector MS Orbitrap Exactive plus, Thermofisher, Massachusetts, USA) equipped with a reverse phase Acquity UPLC BEH C18 column (1.7 µm x 2.1 × 100 mm) at 35 °C.

Chromatographic separation was carried out using a mobile phase consisting of A (acetonitrile and formic acid 0.1%) and B (water and formic acid 0.1%) in a gradient elution analysis programmed as follows: 80% A–20% B for 6 min, then 15% A–85% B for 18 min and 80% A–20% B was maintained for 30 min, at a flow rate of 0.2 mL/min. Calibration curves were obtained by plotting peak areas versus nine different concentrations of standard solutions between 0.05 and 1.5 µg/L. Analyses were performed in triplicate and results were expressed in µg of the specific polyphenol. A good fit (R^2^ > 0.999) was found in the given concentration range (Table 1) with a relatively low limit of detection (LOD: 0.03 µg/L) and limit of quantification (LOQ: 0.1 µg/L) for all of the quantified polyphenols. The relative standard deviation (RSD, %) was taken as a measure of precision for quantitative determination of nine components. 

### 2.10. Computational Chemistry Calculations

The calculations were carried out using the Gaussian 09 [53] program package. The solvation model based on density (SMD) [54] was employed. Following a gas-phase optimization at density functional theory (DFT) M062x/6-311+G(d,p) level of theory, the solvation free energy was calculated from the single-point energy difference between the gas phase and the liquid phase. No imaginary vibrational frequencies were found at the optimized geometries, which indicates that they are true minima of the potential energy surface.

With the aim of incorporating specific intermolecular interactions, such as hydrogen bonds, within our quantum mechanics calculations, we used a discrete-continuum approach. Four solvent explicit molecules were incorporated representing the first solvation sphere, while the continuum model solvent (SMD) was represented varying the dielectric constant value. For the system containing 25%/75% ethanol/water, one ethanol and three water molecules were explicitly considered. While for 50%/50% ethanol/water, two ethanol and two water molecules were explicitly considered. Except for the dielectric constant, the default parameters were used; the weighted average of the values of the dielectric constant of water and ethanol (3:1 and 1:1, respectively) were used to represent the ethanol/water mixtures (25%/75% and 50%/50% respectively) [55]. We compared ΔG_solv_ of chlorogenic and gallic acids in ethanol/water systems with increasing ethanol contents. Specifically, we evaluated the change in the ΔG_solv_ (ΔΔG_solv_) of chlorogenic and gallic acids in a system in which the ethanol/water content was increased from 25%/75% to 50%/50%. 

### 2.11. Statistical Analysis

A factorial experimental design was applied to determine the effect of extraction temperature and co-solvent during extraction on the global recovery of specific polyphenols; this follows the methodology proposed by Mariotti-Celis et al. [39]. In addition, mean and coefficient of variation (CV) results were presented. Analysis of variance (ANOVA) and least significant difference tests were applied to the response variables (p ≤ 0.05). The statistical analyses of data were carried out using the software Statgraphics Plus for Windows 4.0 (Statpoint Technologies, Inc., Virginia, USA). 

## 3. Results and Discussion

### 3.1. Effect of Ethanol as Co-Solvent in HPLE

In order to find the optimal operational conditions in HPLE for the recovery of flavanols, flavonols, stilbenes (resveratrol) and phenolic acids from Carménère pomace, the extraction temperature (90, 120 and 150 °C) and different water–ethanol mixtures (15%, 32.5% and 50%) were assessed, considering the relevant role of the characteristics of the solvent in the extraction of some specific polyphenols.

#### 3.1.1. Phenolic Acids

The yield of phenolic acids in HPLE was at its maximum (85.18 μg/gdp) at the highest temperature and highest ethanol concentration (Table 2 and Figure 1a). García et al. [27] reported that extractability of phenolic acids was enhanced ~8 times when temperature was increased from 100 to 150 °C, using pure water as an extraction solvent. In our study, when temperature is increased from 90 to 150 °C, the extractability of phenolic acids enhanced ~9, ~12 and ~19 times with 15%, 32.5% and 50% of ethanol respectively (Table 2). In HPLE (P: ~10 MPa, T: 150 °C) pure water presents higher polarity (π*: 0.98) than ethanol (π*: 0.27) [22,23]. This difference could explain the higher recovery of phenolic acids with water-ethanol mixtures, in which an ethanol addition would decrease solvent polarity. Phenolic acids are organic molecules which have a greater affinity for organic solvents, such as ethanol, due to intermolecular interactions (dipole-dipole and dispersion forces of London) [14].

The recovery of specific phenolic acids such as gallic, vanillic, caffeic and ferulic increased with temperature and ethanol concentrations. The most extreme HPLE condition (50%, 150 °C) recovered ~12 times more gallic acid compared to lower temperature extractions (Table 2). Grape pomace presents important concentrations of procyanidins, which are galloylated with gallic acid as terminal units [3]. These compounds can be hydrolyzed at high temperatures (T > 120 °C) with the subsequent release of gallic acid in the extracts [19,27,51]. Like the other phenolic acids, chlorogenic acid recovery increased with temperature, but it decreased (from 20% to 69%) with ethanol addition (Table 2). Wijngaard et al. [32] reported similar behavior, arguing that the small size and the high number of carbonyl groups in chlorogenic acid are factors that improve solubility in pure water, when compared to water–ethanol mixtures (25%–75%) at high temperatures (160–193 °C). Chlorogenic acid was the largest phenolic acid analyzed in our extracts. In addition, carbonyl groups are better hydrogen bonding acceptors than hydroxyl groups, and water is a better hydrogen bonding donor than ethanol. Hence, the interaction between the best acceptor and the best donor of hydrogen generates higher stabilization; this could explain that chlorogenic acid, which possesses the highest number of carbonyl groups among the polyphenols studied, decreases its solubility as the ethanol content increases in the water-ethanol mixture.

To verify if the number of carbonyl groups effectively explains the behavior of phenolic acids, we evaluated the ΔΔG_solv_ of chlorogenic and gallic acids in ethanol/water mixtures using chemical quantum calculations. As can be observed in Table 3, when the ethanol content of ethanol/water mixtures increased from 25% to 50%, a lower value of ΔΔG_solv_ was obtained for gallic acid (1.25 kJ/mol) compared to chlorogenic acid (1.52 kJ/mol). These results indicate that gallic acid was better solvated than chlorogenic acid at higher ethanol concentrations, which could explain the tendency observed in the extraction yields of these phenolic acids.

#### 3.1.2. Flavanols

Under subcritical conditions (P: ~10MPa), the highest recovery of flavanols was achieved at the highest temperature (150 °C) and at the intermediate ethanol concentration (32.5%). When temperature was increased from 90 to 150 °C, the flavanols content was enhanced ~25 and ~12 times using ethanol at 15% and 32.5% respectively (Table 2). Similar studies using pure water at high temperatures (from 100 to 200 °C) enhanced the recovery of flavanols by ~2 times [28,36]. 

The recovery of specific flavanols such as catechin, epicatechin and epigallocatechin increased with temperature and ethanol concentrations up to 32.5% (Table 2). Extracts obtained at 32.5%, 150 °C presented a high epigallocatechin relative concentration (~64%) compared to other specific flavanols (Table 2). The skin of grape pomace contains procyanidins rich in epigallocatechin monomers [3] which can be released through hydrolysis reactions in HPLE [51], explaining the high epigallocatechin content in the 150 °C extracts. 

The recovery of flavanols decreased significantly (from 14% to 56%) for ethanol concentrations higher than 32.5% (Table 2 and Figure 1b). Downey et al. [56] also reported that ethanol concentrations higher than 50% decreased the recovery of flavanols, although under atmospheric conditions (P: ~101.3 kPa, T: 23 °C). Figure 1b shows that this effect is more noticeable at higher temperatures. This should be expected since the polarizability (π*) of ethanol decreases significantly (from 0.51 to 0.35) when the temperature increases from 25 °C to 150 °C [12,57].

#### 3.1.3. Flavonols

Extracts obtained at 15%, 150 °C presented ~67% of quercetin and 33% of kaempferol (Table 2). Although high temperatures improved the extraction of flavonols, an increase in the ethanol concentration significantly decreased the recovery of these compounds (from 20% to 80%) (Figure 1b). Wijngaard et al. [32] observed a similar trend in the recovery of flavonols at high concentrations of ethanol (≥ 50%) and high temperatures (≥100 °C) in HPLE. This behavior could be explained by the competitive interactions of ethanol, water and flavonol molecules. The presence of ketone-type carbonyl groups in the structure of flavonols favors their solubility in water. As ethanol concentration increases in the solvent, ethanol-water interactions are more favored than ethanol-flavonol and flavonol-water interactions, which would decrease the solubility of flavonols, reducing their recovery during extraction. 

#### 3.1.4. Stilbenes (Resveratrol)

Extractions at the highest temperature (150 °C) and at the intermediate ethanol concentration (32.5%) maximized the recovery of resveratrol (4.28 μg/gdp), the only stilbene that we quantified. Ethanol concentrations higher than 32.5% significantly decrease its recovery (Table 2 and Figure 1d). This behavior is in agreement with the findings of Karacabey and Mazza [58] who reported that ethanol concentrations higher than 60% decreased the recovery of resveratrol in extractions performed at atmospheric conditions (P: ~101.3 kPa, T: 20–80 °C). Like flavanols, this effect is more pronounced at higher temperatures, and as discussed above, it is probably related to ethanol polarity being reduced at high temperatures.

#### 3.1.5. Interfering Compounds

Ethanol addition (15%–50%) decreased both the glucose and fructose extraction (~60%) (Table 2). Reducing sugars are hydrophilic compounds that easily form hydrogen bonds with water molecules. Therefore, the presence of a low polar co-solvent such as ethanol disfavors their solubility during the extraction [59,60]. In addition, high temperatures (T° ≥ 120 °C) favor the Maillard reaction, where reducing sugars transform into furfural compounds [29].

We observed that ethanol addition reduced the formation of hydroxymethylfurfural (HMF) during HPLE performed at 120 °C (Table 2) which could be attributed to the low content of reducing sugars (main precursors of HMF) in the crude extract. HMF was only detected in extracts obtained at 150 °C (from 11 to 23 mg HMF/gdp) (Table 2). However, this concentration is significantly lower than the levels (80 mg/kg body weight/day) required to observe carcinogenic and genotoxic effects in laboratory animals [61].

#### 3.1.6. Global Antioxidant Properties

The obtained HPLE extracts presented the best global condition at 150 °C with 32.5% of ethanol, in which the total quantified polyphenols families reached ~ 230 µg/gdp. It is in accordance with the total polyphenol content of the extracts which was 50% higher (~54 mg GAE/gdp) than the extracts obtained at the other evaluated conditions. Interestingly, regarding the antioxidant capacity, the highest value was also observed at the same conditions (~340 µM Trolox/gdp).

### 3.2. Purification with Macropoporous Resin (RP)

After HPLE, the extracts were purified using RP to improve the specific separation (free of interfering compounds) of phenolic acids, flavanols, flavonols and stilbenes.

#### 3.2.1. Purification of Phenolic Acids

An increase in the ethanol content of the extracts (15%–50%) decreased the recovery of phenolic acids during RP (Table 4). For example, the extracts obtained at 150 °C with ethanol additions of 15%, 32.5% and 50% presented recoveries over ~65%, ~14% and ~8% of phenolic acids respectively, after RP (Figure 2a). The global HPLE-RP condition, 15%, 150 °C and 80%, allowed the highest recovery of phenolic acids, with gallic acid being the most abundant (~70%) (Table 4). Yang et al. [62] observed a similar behavior during RP, arguing that probably the compounds of interest are more stabilized by the interactions with the solvent than by the interactions with the resin surface. Hence, the higher the ethanol content in the raw extracts, the lower the polyphenols content in the purified extracts (Table 4).

#### 3.2.2. Purification of Flavanols

During RP, the adsorption of flavanols was enhanced significantly when the extracts presented a low ethanol content (Table 4); for example, in this stage, the extracts obtained at 150 °C with ethanol concentrations of 15%, 32.5% and 50% presented recoveries of flavanols of ~57%, ~25% and ~21% respectively (Figure 2b). The global HPLE-RP condition, 15%, 150 °C and 80%, allowed a higher selectivity in the adsorption of epigallocatechin (~56%) compared to the other determined flavanols (Table 4). Mariotti et al. [39] observed a similar behavior during the RP of HPLE extracts obtained at low ethanol concentration (16%). The interactions between the polyphenols, the solvent and the resin would explain the selective recovery observed for epigallocatechin. HP-20 is a polyaromatic adsorbent resin without polar groups. Probably the stability between the van der Waals forces and π–π-stacking interactions, which occur between the aromatic rings of epigallocatechin and the aromatic rings in the surface of the resin, are more stable than the interactions between the ethanol/water mixture and the compound of interest after a determined polarity threshold [39]. 

#### 3.2.3. Purification of Flavonols

The recovery of flavonols after RP was considerably more efficient for the extracts with high ethanol concentrations (Table 4); for example, after RP, the extracts obtained at 150 °C with ethanol concentrations of 15%, 32.5% and 50% presented recoveries of flavonols over ~17%, ~41% and ~61% respectively (Figure 2c). The global HPLE-RP, 50%, 150 °C and 80%, allowed the highest global recovery of quercetin (~62%). The structure of flavonols presents hydroxyl and ketonic groups which improve their stability in polar solvents [63]. High ethanol levels in the raw extracts reduce the relative stability of flavonols in the solvent, favoring their adsorption in the resin surface.

#### 3.2.4. Purification of Stilbenes (Resveratrol)

During RP, the recovery of resveratrol decreased when the ethanol concentration increased in the raw extracts (Table 4). For example, the extracts obtained at 150 °C with ethanol concentrations of 15%, 32.5% and 50% presented recoveries of resveratrol over ~64%, ~19% and ~12% respectively (Figure 2d). Similar to flavanols and phenolic acids, stilbenes are probably more stabilized by the interactions with the solvent than by the interactions with the resin surface, which would explain the observed behavior (Table 4). 

#### 3.2.5. Interfering Compounds

All purified extracts were free from reducing sugars, while HMF was only found in purified extracts obtained at the highest extraction temperature (150 °C). However, it is worth noting that in these extracts HMF was significantly reduced (~95%) by RP (Table 4). The use of non-ionic resins favors interactions with non-polar compounds (e.g., polyphenols), whereas polar compounds (e.g., sugars and HMF) are not adsorbed, facilitating their elimination [35].

## 4. Conclusions

The global recovery of the phenolic acids, flavanols, flavonols and stilbenes quantified in this study in an integrated HPLE-RP processes, significantly increase at 150 °C. On the other hand, the optimal ethanol concentration used in HPLE is specific for each family of polyphenols (flavonols: 15%, flavanols and stilbenes: 32%; and phenolic acids: 50%), but also depends on the chemical structure of the compound of interest. For example, even though total phenolic acid recovery is at a maximum at the highest ethanol concentrations, chlorogenic acid shows lower recoveries at this condition. Using quantum chemical calculations, it was verified that this behavior is due to the high number of carbonyl groups in the compound’s structure, compared to the other phenolic acids considered in this study. The obtained HPLE extracts presented the best global condition at 150 °C with 32.5% of ethanol in which the quantified polyphenol value (~230 µg/gdp), total polyphenol content (~54 mg GAE/gdp) and antioxidant capacity (~340 µM Trolox/gdp) were the highest. The ethanol concentration in HPLE significantly impacts the global yield of the integrated HPLE-RP process. The lower the ethanol content in the HPLE extracts, the higher the global recovery of phenolic acids, flavanols and stilbenes. However, flavonols present the opposite trend, which is attributed to the presence of polar functional groups in their structure. The integrated HPLE/RP separation process eliminates interfering compounds such as glucose and fructose. In addition, this process ensures the safety of the extracts, almost completely removing HMF from all purified extracts.

## Figures and Tables

**Figure 1 molecules-24-03145-f001:**
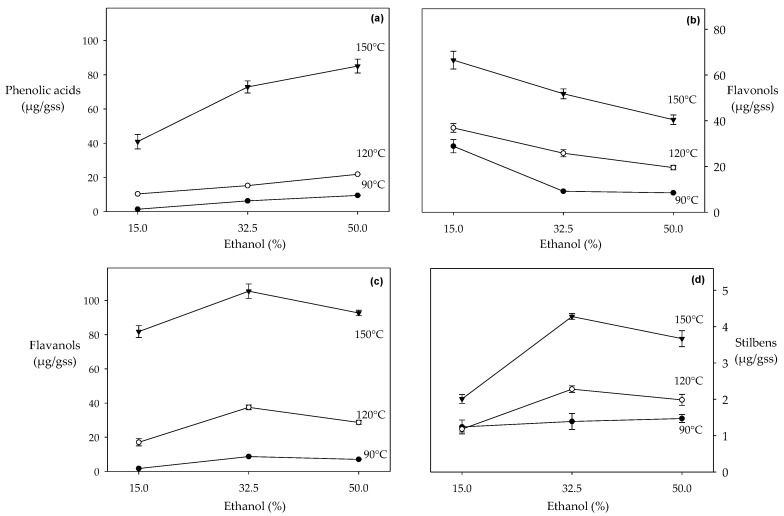
Recovery of polyphenol families in HPLE at different temperatures and ethanol concentrations: (**a**) phenolic acids; (**b**) flavonols; (**c**) flavanols; (**d**) stilbenes (resveratrol).

**Figure 2 molecules-24-03145-f002:**
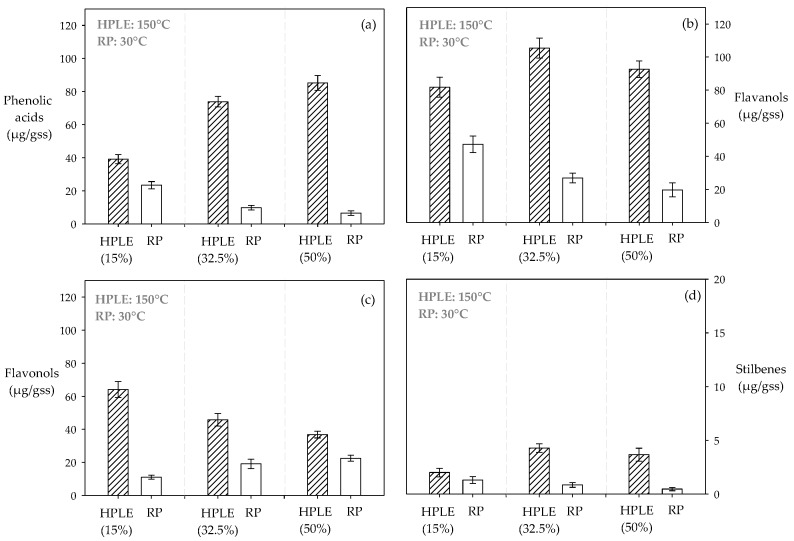
Effect of co-solvent addition on the recovery of polyphenol families in an integrated HPLE-resin purification (RP) process: (**a**) phenolic acids; (**b**) flavanols; (**c**) flavonols; (**d**) stilbenes (resveratrol).

**Table 1 molecules-24-03145-t001:** Analytical features of ultra-performance liquid chromatography coupled to mass spectrometry (UPLC–MS) method for polyphenol quantification in grape pomace extracts.

Target Polyphenols	*m*/*z*	Regression Equation	R^2^
Epigallocatechin	305.066	Y = 4.92436⋅× 10^4^ X	0.9995
Gallic acid	169.015	Y = 1.57814 × 10^6^⋅X	0.9971
Chlorogenic acid	353.087	Y = 3.10744 × 10^6^⋅X	0.9996
Vanillic acid	167.044	Y = 4.12643⋅× 10^4^ X	0.9742
Ferulic acid	193.050	Y = 7.40734⋅× 10^5^ X	0.9999
Catechin	289.071	Y = 3.67625 × 10^6^⋅X	0.9990
Epicatechin	289.071	Y = 4.67949 × 10^6^⋅X	0.9994
Caffeic acid	179.034	Y = 4.54778 × 10^6^⋅X	0.9998
Resveratrol	227.071	Y = 1.12818⋅× 10^5^ X	0.9998
Quercetin	301.035	Y = 1.49192⋅× 10^4^ X	1.0000
Kaempferol	285.040	Y = 2.26042 × 10^6^ X	0.9990

**Table 2 molecules-24-03145-t002:** Polyphenols profile of extracts obtained by hot-pressurized liquid extraction (HPLE) process.

Description	HPLE								
	90 °C			120 °C			150 °C		
	15%	32.5%	50%	15%	32.5%	50%	15%	32.5%	50%
**Acids (µg/gdp)**	Mean CV	Mean CV	Mean CV	Mean CV	Mean CV	Mean CV	Mean CV	Mean CV	Mean CV
Gallic acid	0.97 0.07	3.99 0.05	4.73 0.06	5.80 0.04	10.06 0.04	13.87 0.09	28.54 0.07	56.57 0.09	59.92 0.11
Chlorogenic acid	0.77 0.05	0.23 0.08	0.19 0.04	0.84 0.06	0.39 0.07	0.28 0.07	1.94 0.05	1.36 0.08	1.01 0.05
Vanillic acid	ND	1.57 0.04	3.02 0.05	1.84 0.06	3.50 0.05	6.17 0.08	6.32 0.05	12.99 0.08	20.57 0.06
Caffeic acid	0.22 0.04	0.36 0.03	0.68 0.02	0.69 0.08	0.83 0.05	1.01 0.07	1.80 0.09	2.28 0.06	2.57 0.03
Ferulic acid	ND	0.14 0.07	0.26 0.09	0.30 0.04	0.43 0.08	0.49 0.04	0.53 0.08	0.70 0.08	1.11 0.04
**Σ:**	**1.96 0.05**	**6.29 0.06**	**8.88 0.05**	**9.47 0.06**	**15.21 0.06**	**21.82 0.07**	**39.13 0.06**	**73.90 0.08**	**85.18 0.06**
**Flavanols (µg/gdp)**									
Catechin	0.68 0.04	1.25 0.05	0.94 0.07	4.11 0.05	5.66 0.09	6.06 0.06	15.41 0.08	23.39 0.10	21.05 0.06
Epicatechin	0.71 0.04	1.32 0.11	1.35 0.07	3.22 0.07	5.49 0.06	6.17 0.07	8.50 0.11	14.41 0.09	12.64 0.07
Epigallocatechin	1.77 0.09	6.11 0.06	5.15 0.09	9.76 0.08	26.34 0.11	17.45 0.09	57.89 0.06	67.66 0.10	58.99 0.11
**Σ:**	**3.16 0.06**	**8.68 0.07**	**7.44 0.08**	**17.08 0.07**	**37.51 0.09**	**29.68 0.07**	**81.80 0.08**	**105.46 0.10**	**92.68 0.08**
**Flavonols (µg/gdp)**									
Quercetin	16.97 0.08	6.99 0.09	6.34 0.08	21.40 0.09	14.63 0.07	11.88 0.08	43.14 0.08	34.12 0.10	30.95 0.08
Kaempherol	10.67 0.07	1.08 0.10	0.77 0.07	14.28 0.08	7.39 0.09	5.26 0.07	20.90 0.10	11.72 0.09	5.84 0.09
**Σ:**	**27.64 0.08**	**8.07 0.10**	**7.18 0.08**	**35.68 0.09**	**22.03 0.08**	**17.14 0.08**	**64.04 0.09**	**45.84 0.10**	**36.79 0.09**
**Stilbenes (µg/gdp)**									
Resveratrol	1.24 0.07	1.39 0.09	1.07 0.08	1.18 0.05	2.28 0.06	1.94 0.08	2.01 0.07	4.28 0.08	3.67 0.09
**Interfering (mg/gdp)**									
Glucose	9.85 0.06	7.11 0.07	3.69 0.06	10.36 0.09	8.01 0.06	5.11 0.09	12.63 0.07	10.86 0.05	6.45 0.06
Fructose	7.84 0.08	6.25 0.08	2.94 0.07	9.47 0.08	7.61 0.08	4.50 0.07	11.91 0.04	10.31 0.06	4.87 0.09
HMF	ND	ND	ND	ND	ND	ND	23.61 0.07	17.83 0.05	11.28 0.06

Specific polyphenols content is expressed as µg/g dry pomace. HMF: hydroxymethylfurfural is expressed as mg HMF/g dry pomace. Fructose and glucose contents were expressed as mg/g dry pomace. CV: coefficient of variation. ND: not detected.

**Table 3 molecules-24-03145-t003:** Changes of the Gibbs free energies of solvation of chlorogenic and gallic acids in ethanol-water systems with increasing ethanol contents.

Phenolic Acid	ΔG_solv 25% ethanol_ [kJ/mol]	ΔG_solv 50% ethanol_ [kJ/mol]	ΔΔG_solv_ [kJ/mol]
Gallic acid	−60.66	−59.41	1.25
Chlorogenic acid	−112.51	−110.99	1.52

**Table 4 molecules-24-03145-t004:** Polyphenols profile of purified extracts obtained by integrated HPLE-RP process.

HPLE	90 °C			120 °C			150 °C		
	15%	32.5%	50%	15%	32.5%	50%	15%	32.5%	50%
RP	80%	80%	80%	80%	80%	80%	80%	80%	80%
**Acids (µg/gdp)**	Mean CV	Mean CV	Mean CV	Mean CV	Mean CV	Mean CV	Mean CV	Mean CV	Mean CV
Gallic acid	0.55 0.04	0.30 0.04	0.27 0.05	3.52 0.03	1.61 0.09	0.74 0.07	18.97 0.06	6.21 0.08	3.79 0.07
Chlorogenic acid	0.02 0.02	0.03 0.05	0.07 0.04	0.03 0.02	0.07 0.04	0.05 0.04	0.15 0.02	0.64 0.02	0.86 0.02
Vanillic acid	ND	2.65 0.06	0.58 0.06	1.48 0.06	0.66 0.08	0.35 0.09	4.64 0.09	2.05 0.11	1.54 0.09
Caffeic acid	0.13 0.05	0.14 0.03	0.08 0.03	0.60 0.04	0.57 0.03	0.36 0.04	1.24 0.06	0.61 0.06	0.20 0.06
Ferulic acid	ND	0.06 0.02	0.03 0.02	0.17 0.03	0.12 0.03	0.09 0.05	0.34 0.05	0.26 0.06	0.09 0.05
**Σ:**	**0.68 0.04**	**3.18 0.04**	**1.03 0.04**	**5.80 0.04**	**3.03 0.06**	**1.57 0.06**	**25.34 0.06**	**9.77 0.07**	**6.48 0.06**
**Flavanols (µg/gdp)**									
Catechin	0.52 0.05	0.35 0.08	0.22 0.07	3.17 0.09	1.30 0.09	0.91 0.05	11.42 0.08	8.26 0.06	5.48 0.07
Epicatechin	0.63 0.06	0.44 0.05	0.21 0.08	2.26 0.08	1.51 0.06	0.88 0.07	6.38 0.09	3.84 0.09	2.79 0.08
Epigallocatechin	0.82 0.03	0.52 0.10	0.24 0.09	4.21 0.10	2.27 0.08	1.05 0.09	29.40 0.10	14.75 0.08	11.40 0.07
**Σ:**	**1.97 0.05**	**1.31 0.08**	**0.67 0.08**	**9.64 0.09**	**5.08 0.08**	**2.83 0.07**	**47.20 0.09**	**26.85 0.08**	**19.67 0.07**
**Flavonols (µg/gdp)**									
Quercetin	0.65 0.07	2.81 0.09	3.99 0.09	0.98 0.09	6.56 0.08	8.16 0.10	9.86 0.09	15.28 0.08	19.09 0.08
Kaempherol	0.26 0.06	0.45 0.07	0.47 0.05	0.76 0.05	1.79 0.06	2.73 0.05	1.17 0.05	3.87 0.05	3.47 0.09
**Σ:**	**0.91 0.07**	**3.26 0.08**	**4.46 0.07**	**1.74 0.07**	**8.35 0.07**	**10.89 0.08**	**11.03 0.07**	**19.15 0.07**	**22.56 0.09**
**Stilbenes (µg/gdp)**									
Resveratrol	0.69 0.05	0.42 0.06	0.13 0.06	0.88 0.05	0.70 0.04	0.68 0.06	1.30 0.05	0.85 0.06	0.47 0.05
**Interfering (mg/gdp)**									
Glucose	ND	ND	ND	ND	ND	ND	ND	ND	ND
Fructose	ND	ND	ND	ND	ND	ND	ND	ND	ND
HMF	ND	ND	ND	ND	ND	ND	0.19 0.09	0.13 0.08	0.22 0.07

Specific polyphenols content is expressed as µg/g dry pomace. HMF: hydroxymethylfurfural is expressed as mg HMF/g dry pomace. CV: coefficient of variation. ND: not detected.

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
