# Peer review of "The Impact of Temperature and Ethanol Concentration on the Global Recovery of Specific Polyphenols in an Integrated HPLE/RP Process on Carménère Pomace Extracts"

_molecules, 2019, doi:10.3390/molecules24173145_

Round 1
Reviewer 1 Report
The manuscript entitled: „ The impact of temperature (90 – 150°C) and ethanol concentration (15 – 50%) on the global recovery of specific polyphenols in an integrated HPLE/RP process„ presents some/part of new information, while the study it seems to be well planned, prepared and described. They are only a few small issues which I would like to highlight (please see the comments below).
1. From the conclusion can be understand that each family of polyphenols has specific for requires for the optimal extraction, it would important to provide the information about the compromise parameters which would allow for the relative high extractability of all polyphenols.
2. Since from the stilbenes, only the resveratrol was determined, I would advise to use resveratrol instead of talking about the stilbenes. Additionally, I would aks for updating information about the detailed composition of stilbenes in grape skins, seeds, juice, and stems. The publication listed below should be helpful.
"Identification and determination of stilbenes by Q-TOF in grape skins, seeds, juice, and stems. Journal of Food Composition and Analysis, 2018, 74, 44-52"
Author Response
Recommendation 1.
From the conclusion can be understand that each family of polyphenols has specific for requires for the optimal extraction, it would important to provide the information about the compromise parameters which would allow for the relative high extractability of all polyphenols.
Response: The required information was included in the corrected version of the document. We mentioned in the corrected document (results and conclusions) the best overall conditions, defined as those in which the highest total specific families of polyphenols concentrations were reached. These same conditions allowed us to obtain both the highest total polyphenols content and antioxidant capacity. Please see lines 416-419.
Recommendation 2:
Since from the stilbenes, only the resveratrol was determined, I would advise to use resveratrol instead of talking about the stilbenes. Additionally, I would ask for updating information about the detailed composition of stilbenes in grape skins, seeds, juice, and stems. The publication listed below should be helpful.
"Identification and determination of stilbenes by Q-TOF in grape skins, seeds, juice, and stems. Journal of Food Composition and Analysis, 2018, 74, 44-52".
Response: We included the recommended reference about resveratrol (article: Identification and determination of stilbenes by Q-TOF in grape skins, seeds, juice, and stems). Please see Lines 50-51. In addition, we mentioned that resveratrol was the only stilbene determined (Please see lines: 323-324).
Reviewer 2 Report
The authors study the impact of temperature and ethanol concentration on the global recovery of specific polyphenols in an integrated HPLE/RP process. The manuscript is well written and present, it is recommended for publication after some minor correction.
I think the title should be change to something like
The impact of temperature and ethanol concentration on the global recovery of specific polyphenols in an integrated HPLE/RP process on Carménère pomace extracts
Since the process was optimize only for Carménère pomace, the title should be more specific
I think 78 reference is too many for this manuscript, also the introduction is too long and some parts such as L96-117 and L132-139, I think it can be shortened.
L143 HPL or HPLC?
L151 Authors should provide the purity of all standards
L204 LOD…..0.03__space__ug….
The LOQ should be at least 3.33x of LOD so it cannot be 0.05 ug/L
Author Response
All your suggestions were considered, please see below the specific answer for each one.
Recommendation 1:
The authors study the impact of temperature and ethanol concentration on the global recovery of specific polyphenols in an integrated HPLE/RP process. The manuscript is well written and present, it is recommended for publication after some minor correction.
I think the title should be change to something like
The impact of temperature and ethanol concentration on the global recovery of specific polyphenols in an integrated HPLE/RP process on Carménère pomace extracts
Since the process was optimize only for Carménère pomace, the title should be more specific.
Response: It was done; please see the new title: “The impact of temperature and ethanol concentration on the global recovery of specific polyphenols in an integrated HPLE/RP process on Carménère pomace extracts”
Recommendation 2:
I think 78 reference is too many for this manuscript, also the introduction is too long and some parts such as L96-117 and L132-139, I think it can be shortened.
Response: According to your comment, the number of references was reduced from 78 to 64, including in the corrected document only the ones which contain the most relevant information for the work (Please see the references of the corrected article). Regarding the of length of the introduction, we eliminated some examples according to your suggestion because this information can be accessed through the references of the article.
Recommendation 3:
L143 HPL or HPLC?.
Response: It was corrected to HPLE. Please see line 139.
Recommendation 4:
L151 Authors should provide the purity of all standards
Response: We included the purity of all the standards. This information can be verified in lines 148 – 150 of the corrected manuscript.
Recommendation 5:
L204 LOD…..0.03__space__ug….
Response: It was corrected. Please see lines 214
The LOQ should be at least 3.33x of LOD so it cannot be 0.05 ug/L
Response: It was changed and the LOQ was corrected to 0.1 ug /L please see line 214.